# Experimental Infection of Newly Hatched Domestic Ducklings via Japanese Encephalitis Virus-Infected Mosquitoes

**DOI:** 10.3390/pathogens9050371

**Published:** 2020-05-12

**Authors:** Di Di, Chenxi Li, Junjie Zhang, Muddassar Hameed, Xin Wang, Qiqi Xia, Hui Li, Shumin Xi, Zongjie Li, Ke Liu, Beibei Li, Donghua Shao, Yafeng Qiu, Jianchao Wei, Zhiyong Ma

**Affiliations:** Shanghai Veterinary Research Institute, Chinese Academy of Agricultural Science, Shanghai 200241, China; didi950713@outlook.com (D.D.); lichenxihsy@outlook.com (C.L.); zhangjunjie19970510@outlook.com (J.Z.); mudasar386@gmail.com (M.H.); wangxin1655609668@gmail.com (X.W.); xiaqiqi1996@outlook.com (Q.X.); lihui022715@outlook.com (H.L.); xishumin123@outlook.com (S.X.); lizongjie@shvri.ac.cn (Z.L.); liuke@shvri.ac.cn (K.L.); lbb@shvri.ac.cn (B.L.); shaodonghua@shvri.ac.cn (D.S.); yafengq@shvri.ac.cn (Y.Q.)

**Keywords:** Japanese encephalitis virus, mosquito, domestic duckling, experimental infection, pathogenicity

## Abstract

Japanese encephalitis virus (JEV) is a zoonotic pathogen that is maintained by mosquito vectors and vertebrate hosts including birds in a natural transmission cycle. Domestic ducklings are sensitive to JEV infection, but the clinical responses of domestic ducklings to natural JEV infection are unknown. In this study, we simulated the natural JEV infection of domestic ducklings via JEV-infected mosquito bites to evaluate the pathogenicity of JEV in domestic ducklings. Specific pathogen-free domestic ducklings were infected at day 2 post-hatching with JEV-infected *Culex pipiens* mosquito bites and monitored for clinical responses. Among 20 ducklings exposed to JEV-infected mosquitoes, six showed mild and non-characteristic clinical signs starting at two days post-infection, then died suddenly with neurological signs of opisthotonos (a condition of spasm of the back muscles causing the head and limbs to bend backward and the trunk to arch forward) between two and three days post-infection. The mortality of the affected ducklings was 30% (6/20). Multifocal lymphohistiocytic perivascular cuffs and lymphohistiocytic meningitis were macroscopically observed in the affected duckling brains. JEV was detected in the cytoplasm of neuronal cells in the affected duckling brains by immunohistochemical assays and was recovered from the affected duckling brains by viral isolation. These observations indicated that JEV infection via mosquito bites causes mortality associated with viral encephalitis in newly hatched domestic ducklings, thus demonstrating the potential pathogenicity of JEV in domestic ducklings under natural conditions.

## 1. Introduction

Japanese encephalitis virus (JEV) is a member of the genus *Flavivirus* in the family *Flaviviridae*, which comprises more than 70 species including West Nile virus, Zika virus (ZIKV), and dengue virus [1]. As a zoonotic flavivirus, it is transmitted mainly by mosquito vectors from vertebrate-amplifying hosts to susceptible hosts, and it causes encephalitis in humans, horses, and piglets and abortion and orchitis in breeding pigs [2,3,4]. Birds and pigs are the vertebrate-amplifying hosts, which play essential roles in the bird-associated wild transmission cycle and the pig-associated rural domestic transmission cycle, respectively [5]. In the bird-associated wild transmission cycle, after JEV infection, birds develop a level of viremia sufficient to infect mosquitoes and therefore play an essential role in the maintenance and transmission of JEV in nature [5,6]. More than 90 species including domestic and wild birds are sensitive to JEV infection and develop a variable degree of viremia: some species show subclinical infection, whereas others exhibit clinical signs and even death [7,8,9].

Domestic duck farming is a popular business. Similar to other species of birds, domestic ducks are sensitive to JEV infection [10]. Serological surveys have indicated that 10%, 26%, and 43% of ducks are seropositive for JEV antibodies in India [11], Nepal [12], and Cambodia [13], respectively. The susceptibility of domestic ducks to JEV infection is age-related. Adult domestic ducks are generally asymptomatic, but newly hatched domestic ducklings may develop clinical signs [9] and even death [14] after experimental infection with JEV, thus suggesting the potential pathogenicity of JEV to domestic ducklings. However, these previous experimental infections were performed by subcutaneous injection of ducklings with 10,000–25,000 plaque-forming units (PFU) of JEV [9,14], and the results therefore may not represent the clinical responses to natural infection caused by JEV-infected mosquito bites.

In natural infection, mosquitoes, such as *Culex (Cx.) tritaeniorhynchus* (the primary vector of JEV) and *Cx. pipiens* (the important secondary or regional vector of JEV) [15], acquire JEV by sucking the blood of a viremic host and subsequently become infected. The JEV replicates in the infected mosquitoes, and the progeny viruses disseminate into the salivary gland during a 7- to 14-day extrinsic incubation period. The progeny viruses in the saliva are transmitted to a susceptible host when the infected mosquitoes feed on the host [2]. In this study, we used JEV-infected *Cx. pipiens* mosquitoes to simulate the natural infection of newly hatched domestic ducklings via bites to evaluate the potential pathogenicity of JEV in domestic ducklings.

## 2. Results

### 2.1. Viral Loads in JEV-Infected Cx. Pipiens Mosquitoes

To infect ducklings via JEV-infected mosquito bites, we first prepared JEV-infected *Cx. pipiens* mosquitoes by intrathoracic injection of JEV. During the extrinsic incubation period, five mosquitoes were randomly sampled for detection of viral loads in both whole mosquitoes and secondary organs by quantitative real-time reverse transcription-polymerase chain reaction (qRT-PCR). Analysis of the whole mosquitoes revealed that all five JEV-infected mosquitoes sampled were positive for JEV at 10 days post infection (dpi), with an average of 6.4 × 10^4^
*JEV E gene* copies per mosquito, whereas no JEV was detectable in mock-infected mosquitoes (Figure 1A). Analysis of the secondary organs suggested that JEV disseminated to the secondary organs including salivary glands, limbs, head, and chest, with variable viral loads (Figure 1B). Notably, JEV was detectable in the salivary glands, with a 100% dissemination rate at both 5 and 10 dpi. The viral load in the salivary glands at 10 dpi was 4.9 × 10^4^ copies per mosquito, a value similar to the number of copies of ZIKV found in mosquito saliva in a macaque challenge [16]. We therefore considered the JEV-infected *Cx. pipiens* mosquitoes suitable for infecting ducklings via bites.

### 2.2. Clinical Signs and RNAemia

Among 20 ducklings exposed to JEV-infected *Cx. pipiens* mosquitoes, six showed mild and non-characteristic clinical signs such as vague, anorexia, and reduced water consumption, starting from 2 dpi. The ducklings died suddenly, with neurological signs of opisthotonos (a condition of spasm of the back muscles causing the head and limbs to bend backward and the trunk to arch forward) between 2 and 3 dpi. The remaining ducklings exposed to JEV-infected mosquitoes showed no visible clinical signs (Appendix A). The mortality of the affected ducklings was 30% (6/20) (Figure 2A). RNAemia was detectable in the blood samples from 2 to 5 dpi, with variable levels in the ducklings exposed to JEV-infected mosquitoes (Figure 2B, Appendix A). No clinical signs, death or RNAemia was observed in the ducklings exposed to mock-infected mosquitoes. These data indicated that the infection of ducklings via the route of JEV-infected mosquito bites caused clinical signs, death, and RNAemia in the affected ducklings.

### 2.3. Pathological Lesions

Three ducklings that died at 3 dpi in the group exposed to JEV-infected mosquitoes were examined for pathological lesions. In these ducklings, compared with those exposed to mock-infected mosquitoes, no gross lesions were visible in most organs, with the exception of the brain and spleen. In these ducklings, the brains were swollen, with congested blood vessels on the surface (Figure 3A, JEV), as compared with those from the ducklings exposed to mock-infected mosquitoes (Figure 3A, MOCK). The spleens were enlarged and relatively darker (Figure 3B, JEV) than those from the ducklings exposed to mock-infected mosquitoes (Figure 3B, MOCK). The average spleen weight of the ducklings exposed to JEV-infected mosquitoes was significantly higher than that of the ducklings exposed to mock-infected mosquitoes (Appendix A).

Macroscopically, multifocal lymphohistiocytic perivascular cuffs [17,18] and lymphohistiocytic meningitis [18] were observed in the brains collected from ducklings exposed to JEV-infected mosquitoes (Figure 3C, JEV) but not mock-infected mosquitoes (Figure 3C, MOCK), thus suggesting the presence of viral encephalitis in the affected ducklings. In the spleens collected from ducklings exposed to JEV-infected mosquitoes, as compared with JEV-infected mosquitoes, widened white pulp and atrophied red pulp were observed (Appendix A, JEV). In addition, fewer splenic cells were observed, some of which were deformed (Appendix A, JEV).

### 2.4. Detection of JEV in Brains

Given the presence of JEV encephalitis in the affected duckling brains, we examined JEV in the brains by immunohistochemical assays with antibodies specific to the JEV NS3 protein. The NS3 protein, shown as brown deposits in the cytoplasm by immunoperoxidase staining, was detectable in the cytoplasm of neuronal cells (marked with red arrows) in the brain sections (Figure 4A, JEV) from the ducklings exposed to JEV-infected mosquitoes but not mock-infected mosquitoes (Figure 4A, MOCK), thus demonstrating the presence of JEV in the affected duckling brains. To confirm this result, the brains were further examined by RT-PCR to detect the presence of the *JEV E gene*. As shown in Figure 4B, a band corresponding to the amplified size of the *JEV E gene* fragment was detected in the brains from all three ducklings exposed to JEV-infected mosquitoes but not mock-infected mosquitoes. Additionally, the viral loads in the brains and other tissues from the affected ducklings were determined by qRT-PCR. The *JEV E gene* was detectable in all tissues examined, with the highest copies in the spleens followed by the brains (Figure 4C,D).

### 2.5. Isolation of JEV From the Affected Duckling Brains

To recover JEV from the affected duckling brains, we collected brain samples from each duckling and inoculated them on baby hamster kidney (BHK) cells. The JEV-characteristic cytopathic effects (CPE) appeared at 3–4 dpi in all BHK cells inoculated with the brain samples collected from the ducklings exposed to JEV-infected mosquitoes (Figure 5A, JEV) but not mock-infected mosquitoes (Figure 5A, MOCK). Total RNA was extracted from the BHK cells and subjected to *JEV E gene* detection by RT-PCR. A specific band corresponding to the amplified size of the *JEV E gene* fragment was detected in the cells with CPE (Figure 5B, JEV) but not in the cells without CPE (Figure 5B, MOCK). The amplified bands were purified from the gel and subjected to DNA sequencing. The resulting nucleotide sequence was compared with the sequence of SH7 strains used for infection of mosquitoes. The nucleotide sequence obtained from the cells (Appendix A) showing CPE was identical to that of the SH7 strain. The supernatants of the cells with and without CPE were collected and subsequently used to inoculate BHK cells; immunofluorescence assays for the presence of JEV were then performed with anti-NS3 antibodies. As shown in Figure 5C, NS3-positive cells were detectable in the cells inoculated with the supernatants from the cells that showed CPE (Figure 5C, JEV) but not in those inoculated with the supernatants from the cells that did not show CPE (Figure 5C, MOCK). Together, these data indicated that the inoculated JEV was recovered from the ducklings exposed to JEV-infected mosquitoes. Additionally, the recovered JEV was sequenced, and no mutation was observed in both nucleotide and amino acid sequences as compared with the SH7 strain (Appendix A).

## 3. Discussion

Experimental infection of newly hatched domestic ducklings via subcutaneous injection of JEV results in stunted growth and death, thus suggesting the potential pathogenicity of JEV in domestic ducklings [9,14]. However, the clinical responses of domestic ducklings to natural JEV infection caused by mosquito bites are unknown. Because domestic duck farming is a popular business, understanding the effects of natural JEV infection on domestic ducklings would benefit domestic duck farming. In this study, to evaluate the potential pathogenicity of JEV, we stimulated natural infection by exposing domestic ducklings to JEV-infected *Cx. pipiens* mosquitoes.

The *Cx. pipiens* mosquito is an important secondary or regional vector of JEV [15] that feeds primarily on birds [19,20]. Therefore, we used JEV-infected *Cx. pipiens* mosquitoes to infect domestic ducklings. Infection of domestic ducklings at day 2 post-hatching via JEV-infected *Cx. pipiens* mosquito bites resulted in mild and non-characteristic clinical signs and caused 30% (6/20) mortality, in agreement with our previous observations [14]. Notably, the affected ducklings died suddenly, with neurological signs of opisthotonos. Gross lesions were observed mainly in the brains, which were swollen with congested blood vessels on the surface, and in the spleens, which were enlarged and relatively darkened. Histopathological examination indicated multifocal lymphohistiocytic perivascular cuffs and lymphohistiocytic meningitis in the brain sections. Multifocal lymphohistiocytic perivascular cuffs are the lesion of JEV encephalitis, as observed in the brains of JEV-infected macaques [17], pigs [3,18], and mice [21]. In addition, lymphohistiocytic meningitis is also observed in JEV-infected pigs [18]. The inoculated JEV was detectable in the cytoplasm of the neuronal cells of the affected duckling brains and was recovered from the affected duckling brains by viral isolation. Overall, these observations indicated that JEV infection via mosquito bites causes mortality associated with viral encephalitis in newly hatched domestic ducklings.

The mortality associated with viral encephalitis observed in this study was not seen in the domestic ducklings subcutaneously inoculated with the same JEV strain used in our previous study [14], in which subcutaneous infection with the same JEV strain did not cause death in ducklings at the same age. This apparent difference in the pathogenicity of the same JEV strain may be attributable to the difference in the inoculation route between the two experimental challenges. In the previous study, the ducklings were inoculated via subcutaneous injection [14], whereas in the present study, the ducklings were inoculated via JEV-infected mosquito bites. The components of mosquito saliva play roles in modulating host immune responses and in facilitating the replication and transmission of flavivirus. For example, saliva and salivary gland extracts from *Cx. tarsalis* mosquitoes enhance the replication of West Nile virus in mice [22]. The salivary factor LTRIN from *Aedes aegypti* mosquitoes facilitates the transmission of ZIKV and exacerbates its pathogenicity by interfering with the lymphotoxin-β receptor in mice [23]. The saliva-specific protein AaVA-1 from *Aedes aegypti* mosquitoes promotes dengue and Zika virus transmission by activating autophagy in host immune cells of the monocyte lineage [24]. According to these previous findings in other flavivirus species, we speculate that the non-identified component(s) from the saliva of JEV-infected *Cx. pipiens* mosquitoes might modulate the duckling immune response and facilitate JEV replication, thereby enhancing the pathogenicity of the JEV strain in ducklings. However, this speculation must be tested in comprehensive experiments.

Although we demonstrated the potential pathogenicity of JEV in newly hatched ducklings, death associated with natural JEV infection is not observed in duck farms. The clinical response and susceptibility of domestic ducklings to JEV infection are known to be age-related, and the clinical signs appear only in ducklings experimentally infected at several days post-hatching [9]. In JEV epidemic regions, 10%–43% ducks are serologically positive for JEV antibodies [11,12,13]. The maternal antibodies, which are passed from the breeder through the egg to their offspring to provide protection against pathogens in the early stages of their offspring’s life, may reduce the clinical response and susceptibility of newly hatched ducklings to JEV infection, thus potentially explaining why no death associated with natural JEV infection has been observed in duck farms. In addition, the clinical signs of the ducklings were mild and non-characteristic, and the affected ducklings died suddenly with neurological signs of opisthotonos, a finding relatively similar to the clinical signs of duck viral hepatitis, the major disease affecting newly hatched ducklings [25]. Death associated with natural JEV infection in domestic ducklings might potentially be misdiagnosed or ignored [9]. Detection of JEV in sick ducklings during mosquito season would clarify the potential pathogenicity of JEV in domestic ducklings.

In conclusion, we simulated natural JEV infection via JEV-infected *Cx. pipiens* mosquito bites to evaluate the potential pathogenicity of JEV in newly hatched domestic ducklings. Infection of domestic ducklings at day 2 post-hatching via JEV-infected mosquito bites caused 30% mortality, which was associated with viral encephalitis. These observations demonstrated the potential pathogenicity of JEV in domestic ducklings under natural conditions. Surveillance of JEV infection in sick ducklings during mosquito season is recommended.

## 4. Materials and Methods 

### 4.1. Ethics Statement

All animal experiments were approved by the Institutional Animal Care and Use Committee of Shanghai Veterinary Research Institute (IACUC No: SHVRI-SZ-2019070603) and were performed in compliance with the Guidelines on the Humane Treatment of Laboratory Animals (Ministry of Science and Technology of the People’s Republic of China, Policy No. 2006 398).

### 4.2. Virus and Cells

The JEV SH7 strain (GenBank No. MH753129) previously isolated from mosquitoes in 2016 [14] was grown and titrated in BHK cells. The titrated JEV stock was stored at -80 ˚C until use. BHK cells were cultured in Dulbecco’s modified eagle medium (DMEM) containing 10% fetal bovine serum (Thermo Fisher Scientific, Waltham, MA, USA), 100 μg/mL streptomycin, and 100 IU/mL penicillin.

### 4.3. Mosquito Rearing and Infection

*Cx. pipiens* mosquitoes were provided by Dr. Zhu Huiman from the Second Military Medical University Shanghai, PR China, and reared as described previously [26]. For JEV infection, 6-day-old female *Cx. pipiens* mosquitoes with similar body size were anesthetized on ice and intrathoracically injected with 100 PFU JEV (0.5 μL per mosquito) with an Eppendorf CellTram oil microinjector (Eppendorf, Hamburg, Germany), as described previously [21]. A mock-infected group of mosquitoes was intrathoracically injected with the same amount of DMEM. The mosquitoes were intrathoracically infected to avoid differences in the JEV quantities at the beginning of infection [27,28]. The infected mosquitoes were held for an extrinsic incubation period and randomly sampled at 5 and 10 dpi for detection of JEV infection and dissemination, by qRT-PCR.

### 4.4. Infection of Ducklings by Mosquito Bites

A schematic representation of the infection of ducklings by JEV-infected mosquito bites is shown in Figure 6. Two-day-old specific pathogen-free Shaoxing ducklings were randomly divided into JEV- and mock-inoculated groups. The JEV- and mock-infected mosquitoes were prevented from eating and drinking by removing of sugar and water from 9 to 10 dpi, then immediately given access to the ducklings. The ducklings together with the starved mosquitoes were placed in a mosquito cage to allow biting. Each mosquito cage contained one duckling and five starved mosquitoes and was kept in the dark for 2 h at 27 °C and 85% relative humidity. After mosquito bites were completed, the status of blood engorgement of each mosquito was visually determined. The clinical responses of the ducklings from the mosquito cages containing at least one blood-engorged mosquito were monitored daily for seven days. Blood samples were collected daily from 2–5 dpi for detection of RNAemia.

### 4.5. Pathological and Immunohistochemical Assays

Three ducklings that died at 3 dpi in the group exposed to JEV-infected mosquitoes were examined for gross lesions. In parallel, three ducklings from the group exposed to mock-infected mosquitoes were euthanized at 3 dpi as mock-infected controls. Tissue samples were collected from the ducklings and subjected to histopathological and immunohistochemical assays. For histopathological assays, the tissue sections were stained with hematoxylin and eosin. For immunohistochemical assays, the tissue sections were probed with antibodies specific to JEV NS3 protein, as described previously [29].

### 4.6. Detection of JEV by qRT-PCR and RT-PCR

Copies of the *JEV E gene* in samples collected from mosquitoes and ducklings were quantified by qRT-PCR with the primers 5′-CATGTGARGACAATCA-3′ and 5′-CCAGCAAGCCTTTTTR-3′ [30]. Briefly, total RNA was extracted from the samples with TRIzol reagent (Thermo Fisher Scientific, Carlsbad, CA, USA) and reverse-transcribed with a PrimeScript™ RT Reagent Kit with gDNA Eraser (TaKaRa, Liaoning, China). The qRT-PCR, standard curve generation, and calculation of JEV E copies were performed as described previously [30]. The qualitative detection of the presence of JEV was performed by RT-PCR with primers targeting the *JEV E gene* (5′-TTGGTCGCTCCGGCTTACA-3′ and 5′-GGTTTTCCGAGGTAGTGGTTC-3′).

### 4.7. Recovery of JEV from Duckling Brains

Brain tissues collected from the ducklings that died at 3 dpi in the group exposed to JEV-infected mosquitoes were homogenized in DMEM using a glass homogenizer and were centrifuged at 2000× *g* at 4 °C for 15 min. The supernatants were diluted with DMEM containing 100 μg/mL streptomycin and 100 IU/mL penicillin and were passed through a 0.38 μm filter, after which the filtrates were inoculated on BHK cells at 37 °C for 2 h. After inoculation, the cells were washed with phosphate-buffered saline and cultured in DMEM containing 2% fetal bovine serum. In parallel, the filtrates prepared from the ducklings exposed to mock-infected mosquitoes were inoculated on BHK cells as mock-infected controls. The cells were monitored daily for CPE of JEV for seven days. The cells showing CPE were subjected to JEV detection by RT-PCR. The products amplified by RT-PCR were sequenced and compared with those from the JEV SH7 strain. The supernatants from cells showing CPE were further inoculated on BHK cells. The cells were incubated for 36 h and subjected to immunofluorescence assays with anti-NS3 antibody, as described previously [29].

## Figures and Tables

**Figure 1 pathogens-09-00371-f001:**
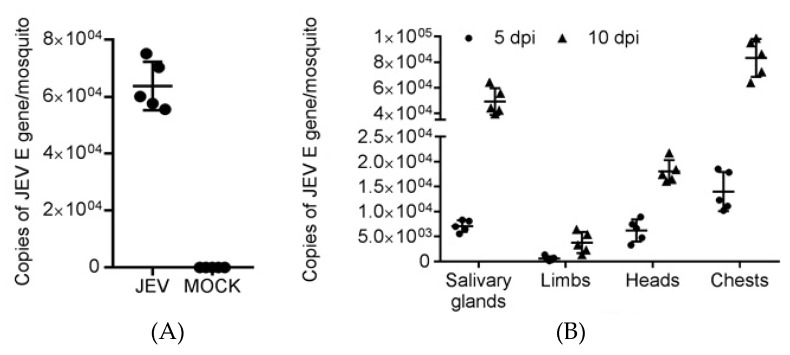
Detection of viral loads in mosquitoes. Female *Cx. pipiens* mosquitoes were intrathoracically mock-infected with DMEM or were infected with JEV at 100 PFU. (**A**) Five mosquitoes from the mock- and JEV-infected groups were randomly sampled at 10 dpi for detection of viral loads in whole mosquitoes by qRT-PCR. (**B**) Five mosquitoes from the mock- and JEV-infected groups were randomly sampled at 5 and 10 dpi, and the secondary organs including the salivary glands, limbs, head, and chest were dissected and subjected to detection of viral loads by qRT-PCR. Each dot represents an individual mosquito.

**Figure 2 pathogens-09-00371-f002:**
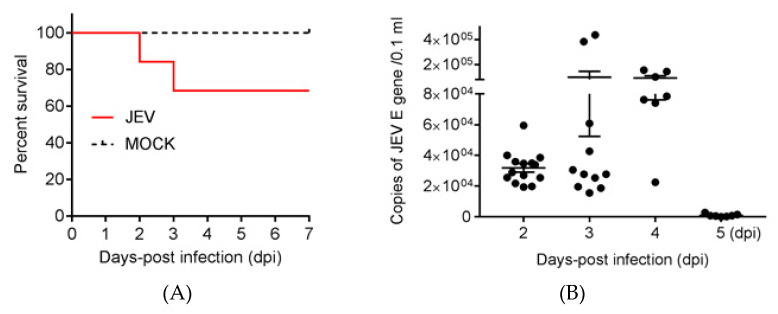
Survival curve and RNAemia. Ducklings (*n*=20) were exposed to mock- and JEV-infected mosquitoes and were monitored daily for seven days. (**A**) Survival curve. (**B**) Blood samples were collected at 2 (*n* = 14), 3 (*n* = 11), 4 (*n* = 7), and 5 dpi (*n* = 7) for analysis of RNAemia by qRT-PCR. Each dot represents an individual duckling.

**Figure 3 pathogens-09-00371-f003:**
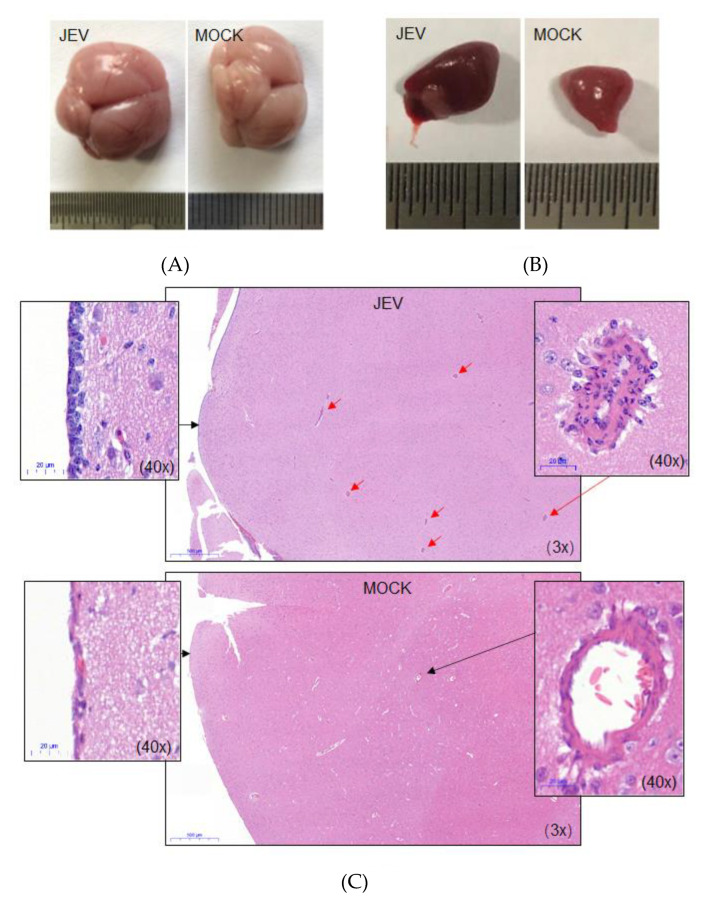
Pathological lesions. Ducklings (JEV) that died at 3 dpi in the group exposed to JEV-infected mosquitoes were examined for pathological lesions. Ducklings (MOCK) from the group exposed to mock-infected mosquitoes were euthanized at 3 dpi as mock-infected controls. (**A**) Gross lesions of the brain. (**B**) Gross lesion of the spleen. (**C**) Histopathological lesions of the brain. Lymphohistiocytic perivascular cuff, indicated by red arrows in the brain sections from the ducklings exposed to JEV-infected mosquitoes.

**Figure 4 pathogens-09-00371-f004:**
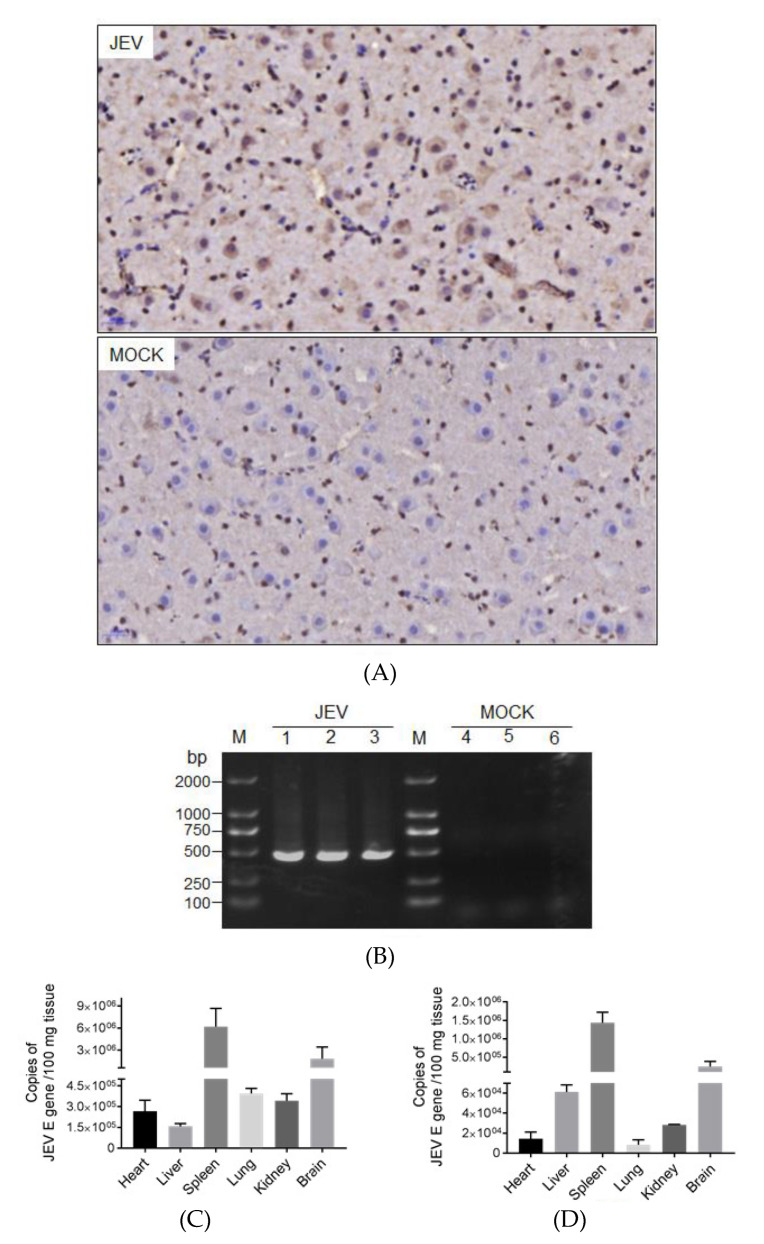
Detection of JEV in the brains and other tissues. Brains (JEV) were collected from the ducklings that died at 3 dpi in the group exposed to JEV-infected mosquitoes, and JEV was detected. Brains (MOCK) from the ducklings exposed to mock-infected mosquitoes were used as mock-infected controls. (**A**) Brains were examined by immunohistochemical assays with anti-NS3 antibodies. The JEV positive cells are indicated by red arrows. 40× magnification. (**B**) RT-PCR detection of JEV in brain samples. Lanes 1–3 are the brain samples from three ducklings exposed to JEV-infected mosquitoes (lanes 1, 2, and 3 represent duckling No. 2, 5, and 15, respectively). Lanes 4–6 are the brain samples from three ducklings exposed to mock-infected mosquitoes. (**C** and **D**) RNAemia in the brains and other tissues collected from affected ducklings No. 5 (**C**) and 15 (**D**) was determined by qRT-PCR.

**Figure 5 pathogens-09-00371-f005:**
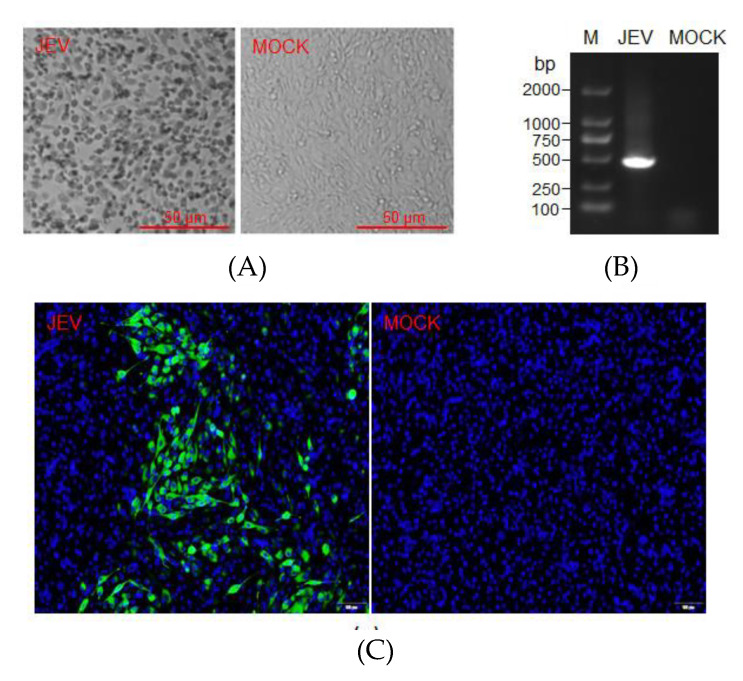
Recovery of JEV from duckling brains. BHK cells were inoculated with brain samples collected from ducklings exposed to JEV- or mock-infected mosquitoes. (**A**) CPE that developed at 3 dpi were photographed. (**B**) RT-PCR detection of JEV in the cells with CPE (JEV) and without CPE (MOCK). (**C**) BHK cells were inoculated with the supernatants from the cells with and without CPE and were immunostained with anti-NS3 antibodies (green) at 36 hpi. The nuclei were stained with DAPI (blue) and were merged with the immunostained images.

**Figure 6 pathogens-09-00371-f006:**
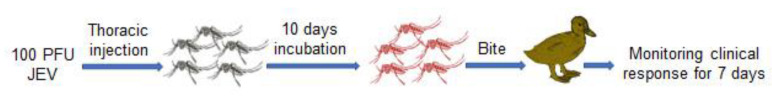
Schematic representation of infection of ducklings by JEV-infected mosquito bites.

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
