# Peer review of "Experimental Infection of Newly Hatched Domestic Ducklings via Japanese Encephalitis Virus-Infected Mosquitoes"

_pathogens, 2020, doi:10.3390/pathogens9050371_

Round 1
Reviewer 1 Report
The manuscript entitled "Experimental infection of newly hatched domestic ducklings via Japanese encephalitis virus-infected mosquito bites” describes a novel JEV infection model mimicking natural exposure. The paper provides interesting data and the overall quality of the manuscript seems fine. Nevertheless, the sample analysis and the presentation of the data should be improved. Furthermore, several details about the study are missing in the M&M section.
Specific comments
- “ Mosquito bites” in the title refers to infected; this seems grammatically not optimal/correct, I propose to change to: Experimental infection of newly hatched domestic ducklings via Japanese encephalitis virus-infected mosquitoes
- The numbering of the figures is a bit strange as figure 1 is presented after de rest of the figures
- In figure 2B the dissemination of the virus in mosquitoes is shown. The variability in virus yield per mosquito is remarkably low. Do the authors always observed such a low variability?
- In figure 3B kinetics of individual ducklings is not shown. It would be informative to present each duckling with a different symbol
- In figure 5A the antigen staining is visible however the staining is not very clear. It seems there is some optimization of the IHC possible?
- In Figure 5B the authors show conventional PCR data which is only qualitative. The authors should show quantitative data of all the organ material derived from the trial.
- In figure 6 the authors confirm the identity of JEV in the brain samples, which is of course good to do, but it does not really add a lot. Whole viral genome sequencing of duck passaged virus would for instance be more interesting.
- It is unclear what the relationship between the number of mosquitos taken a bloodmeal on each duckling and subsequent pathology is.
- M&M section: Homogenized with DMEM? I assume a sort of tissue homogenizer has been used?
Author Response
Responses to the reviewer 1 comments
Point 1: “Mosquito bites” in the title refers to infected; this seems grammatically not optimal/correct, I propose to change to: Experimental infection of newly hatched domestic ducklings via Japanese encephalitis virus-infected mosquitoes
Response 1: Thank you for the suggestion. The title has been changed as per suggestion. Please see the line number 2 in the revised manuscript with track changes.
Point 2: The numbering of the figures is a bit strange as figure 1 is presented after de rest of the figures.
Response 2: Thank you for your kind correction. We have re-numbered the Figures accordingly. Please see the line number 69, 71, 77, 90, 92, 96, 104, 105, 106, 107, 109, 112, 115, 116, 119, 129, 130, 132, 138, 152, 153, 155, 161, 162, 163, 169, 260 and 270 in the revised manuscript with track changes.
Point 3: In figure 2B the dissemination of the virus in mosquitoes is shown. The variability in virus yield per mosquito is remarkably low. Do the authors always observed such a low variability?
Response 3: Thank you for the question. Mosquitoes vary in body size because of their hatching habits, especially in the early life after hatching. In our preliminary trial, high variability in virus yield was observed when we infected mosquitoes with different body sizes, which was not suitable for subsequent infection of ducklings. To reduce the variability of virus load in mosquitoes, we selected the mosquitoes with same age (hatched on same day) and similar body size for JEV infection and finally obtained the JEV-infected mosquitoes with relatively low variability in virus loads. We have re-written the Materials and methods to mention the mosquito body size. Please see the line number 250 in the revised manuscript with track changes.
Point 4: In figure 3B kinetics of individual ducklings is not shown. It would be informative to present each duckling with a different symbol
Response 4: Thank you for the suggestion. The data of individual ducklings have been provided as supplementary data in Table S1 in the revision.
Point 5: In figure 5A the antigen staining is visible however the staining is not very clear. It seems there is some optimization of the IHC possible?
Response 5: Thank you for the suggestion. We optimized the image by slightly increasing the contrast ratio. Please see the Figure 4A. If it is not acceptable, we prefer to use the original image.
Point 6: In Figure 5B the authors show conventional PCR data which is only qualitative. The authors should show quantitative data of all the organ material derived from the trial.
Response 6: Thank you for the suggestion. We had the quantitative data of JEV loads in different tissues from two died ducklings, and we have provided in Figure 4C and 4D in the revision.
Point 7: In figure 6 the authors confirm the identity of JEV in the brain samples, which is of course good to do, but it does not really add a lot. Whole viral genome sequencing of duck passaged virus would for instance be more interesting.
Response 7: Thank for the suggestion. We have sequenced the JEV recovered from the affected duckling brains and found no mutation in the nucleotide and amino acid sequences as compared with the inoculated JEV SH7 strain. We have provided the sequences as supplementary data in Figure S2 in the revision.
Point 8:It is unclear what the relationship between the number of mosquitos taken a bloodmeal on each duckling and subsequent pathology is.
Response 8: Thank you for the question. We observed that the number of blood-engorged mosquitoes after biting varied among different cages, some mosquitoes did not show visible blood-engorgement even after several times of biting. Among 20 ducklings that have been bitten by 1-5 mosquitoes as confirmed by blood-engorgement, 6 ducklings developed clinical signs, however the remaining ducklings that have also been bitten by 1-5 mosquitoes did not show visible clinical signs. RNAemia of JEV were detected in all ducklings after exposing to JEV-infected mosquitoes, suggesting that JEV infection occurred in all ducklings. It seems that the clinical response of ducklings was related to the individual susceptibility and resistance to JEV infection, but not to the number of mosquito bites. We have provided the individual data of all ducklings as supplementary data in Table S1 in the revision.
Point 9: M&M section: Homogenized with DMEM? I assume a sort of tissue homogenizer has been used?
Response 9: Thank you for the question. We have re-written this sentence to avoid any misunderstanding as following: Brain tissues…were homogenized in DMEM using a glass homogenizer. Please see the line number 290 in the revised manuscript with track changes.
Reviewer 2 Report
The manuscript "Experimental infection of newly hatched domestic ducklings via Japanese encephalitis virus-infected mosquito bites" reports the outcome of ducklings infected with JEV (genotype not specified in paper) by mosquito bite and showed a mortality of 30%. This builds on a number of recent studies investigating JEV infection in juvenile poultry. The overall conclusions are supported by the data presented although a number of improvements could be made as listed below:
- The editing of the manuscript needs correction as the figures are not in the correct sequence with Figure 2 being the first encountered and Figure 1 the last. Also the text of the Figure legends is now incorporating into the text of the manuscript suggesting repetition and needs movement back to the Figure legend.
- Abstract - perhaps finish with the phrase "..under natural conditions." Define opisthotonos.
- Introduction- convention is that Flavivirus, Flaviviridae and species names are in italic. Reference 5 is incomplete. Rephrase the sentence "In the bird-associated wild transmission cycle, after infection, birds develop.."
- Results - 2.2-The authors should also note that they have not demonstrated viraemia, only detection of virus genome (copies E gene/0.1mL [blood?]). The relationship between live virus and this surrogate is not established in the paper so caution in the interpretation and wording is needed. 2.3- The cause of the encephalitis has not been established so wording should be "Given the presence of encephalitis...".
- Discussion - Reference 15 is not appropriate. The statement "hallmark of JEV" needs removing. Encephalitis is the hallmark of many virus infections of the brain. Clarify what maternal antibodies are in the context of birds.
- Methods 4.4-clarify the statement "prevented from eating and drinking", presumably access to sugar or water removed 24 hours prior to feeding on ducklings. 4.6- State how genome copies were calculated.
Author Response
Responses to the reviewer 2 comments
Point 1: The editing of the manuscript needs correction as the figures are not in the correct sequence with Figure 2 being the first encountered and Figure 1 the last. Also the text of the Figure legends is now incorporating into the text of the manuscript suggesting repetition and needs movement back to the Figure legend.
Response 1: Thank you for your kind correction. We have re-numbered the Figures accordingly. Please see the line number 69, 71, 77, 90, 92, 96, 104, 105, 106, 107, 109, 112, 115, 116, 119, 129, 130, 132, 138, 152, 153, 155, 161, 162, 163, 169, 260 and 270 in the revised manuscript with track changes.
Point 2: Abstract - perhaps finish with the phrase "..under natural conditions." Define opisthotonos.
Response 2: Thank for the suggestion. We have re-written the last sentence of the Abstract and provided the definition of opisthotonos in the revision as per suggestion. Please see the line number 21 and 29 in the revised manuscript with track changes.
Point 3: Introduction- convention is that Flavivirus, Flaviviridae and species names are in italic. Reference 5 is incomplete. Rephrase the sentence "In the bird-associated wild transmission cycle, after infection, birds develop.."
Response 3: Thank you for the suggestion.
We have changed the Flavivirus, Flaviviridae and species names of mosquitoes into Italic formation in the revision.
We have re-written the sentences accordingly (please see the line 39 in the revised manuscript with track changes).
The reference 5 has been re-cited.
Point 4: Results - 2.2-The authors should also note that they have not demonstrated viraemia, only detection of virus genome (copies E gene/0.1mL [blood?]). The relationship between live virus and this surrogate is not established in the paper so caution in the interpretation and wording is needed. 2.3- The cause of the encephalitis has not been established so wording should be "Given the presence of encephalitis...".
Response 4: Thank you for the correction. We have replaced the viremia with RNAemia and used the presence of encephalitis in the revision. Please see the line number 84, 90, 92, 94, 95, 96, 99, 268 in the revised manuscript with track changes.
Point 5: Discussion - Reference 15 is not appropriate. The statement "hallmark of JEV" needs removing. Encephalitis is the hallmark of many virus infections of the brain. Clarify what maternal antibodies are in the context of birds.
Response 5: Thank for the correction.
Reference 15 was wrongly cited by Endnote. We have re-cited the reference 15 with the correct one. Please see the line number 352 in the revised manuscript with track changes.
The "hallmark of JEV" has been removed in the revision. Please see the line number 24, 111, 192 in the revised manuscript with track changes.
The explanation of maternal antibodies has been provided in the Dissection. Please see the line number 220-221 in the revised manuscript with track changes.
Point 6: Methods 4.4-clarify the statement "prevented from eating and drinking", presumably access to sugar or water removed 24 hours prior to feeding on ducklings. 4.6- State how genome copies were calculated.
Response 6: Thank you for the question.
We have re-written the sentence as following: “The JEV- and mock-infected mosquitoes were prevented from eating and drinking by removing of sugar and water from 9 to 10 dpi”. Please see the line number 262 in the revised manuscript with track changes.
The method for calculating JEV gene copies has been provided in the revision. Please see the line number 283-284 in the revised manuscript with track changes.
Round 2
Reviewer 2 Report
The authors have addressed the questions raised by the reviewers. The latest draft still requires a thorough proof read as the title was not correct (mosquitoes present in the authors section) and the figure legends need revising so that they do not form part of the text.
Author Response
Response to Reviewer 2 Comments
Reviewer 2:
Point 1: The authors have addressed the questions raised by the reviewers. The latest draft still requires a thorough proof read as the title was not correct (mosquitoes present in the authors section) and the figure legends need revising so that they do not form part of the text.
Response 1:
Thank you for the suggestion.
We have re-formed the title as per suggestion.
All figure legends have been re-organized according to the template of the journal.